# Tsunamigenic potential of a Holocene submarine landslide along the North Anatolian Fault (North Aegean Sea, off Thasos Island): insights from numerical modeling

Alexandre JANIN [1], Mathieu RODRIGUEZ [1], Dimitris SAKELLARIOU [2], Vasilis LYKOUSIS [2], and Christian GORINI [3]

[1]Laboratoire de Géologie de l'Ecole normale supérieure de Paris; PSL research university, CNRS UMR 8538, 24 rue Lhomond, 75005 Paris, France.
[2]Institute of Oceanography, Hellenic Center of Marine Research, GR-19013 Anavyssos, Greece
[3]Sorbonne Universités, UPMC Université Paris 06, UMR 7193, ISTeP, F-75005, Paris, France.

*Correspondence to:* Alexandre JANIN (alexandre.janin@ens.fr)

**Abstract.** The North Anatolian Fault in the northern Aegean Sea triggers frequent earthquakes of magnitude up to $M_w \sim 7$. This seismicity can be a source of modest tsunamis for the surrounding coastlines with less than $50$ cm height according to numerical modelling and analysis of tsunami deposits. However, other tsunami sources may be involved, like submarine landslides. We assess the severity of this potential hazard by performing numerical simulations of tsunami generation and propagation from a Holocene landslide ($1.85$ km$^3$ in volume) identified off Thasos island. We use a model coupling the simulation of the submarine landslide, assimilated to a granular flow, to the propagation of the tsunami wave. The results of these simulations show that a tsunami wave of water height between $1.10$ m and $1.65$ m reaches the coastline at Alexandroupolis ($58.000$ inhabitants) one hour after the triggering of the landslide. In the same way, tsunamis waves of water height between $0.80$ m and $2.00$ m reach the coastline of the Athos peninsula $9$ min after the triggering of the landslide. Despite numerous earthquakes of $M_w > 7$ and strong detrital input (on the order of $30$ cm.ka$^{-1}$), only a few Holocene landslides have been recognized so far, asking for tsunami recurrence in this area.

## 1 Introduction

Tsunamis constitute a major natural hazard for coastal populations and infrastructures. Tsunamis result from an impulsive perturbation of the seafloor, which generates waves with a wavelength $\lambda \sim 100$ km and a period in the $10 - 30$ min range. According to the shallow-water approximation, the phase velocity of tsunami waves is expressed by $c = \sqrt{gh}$ at long period, where $h$ is the bathymetry and $g$ the gravitational acceleration. Although most of the deadliest tsunamis in the last decades result from earthquakes nucleated along submarine faults (Okal, 2015), other sources of damaging tsunamis have been identified, including submarine landslides (Tappin et al., 2008; ten Brink et al., 2014). One of the most conspicuous case-study is the July 1998 event that stroke the coastline of Papua-New Guinea. There, a $M_w \sim 7$ earthquake triggered a $\sim 4$ km$^3$ submarine landslide offshore the Sissano Bay. The slide motion generated a tsunami with run-up values locally $> 10$ m, which caused more than 2200 casualties (Heinrich et al., 2000, 2001; Tappin et al., 2001, 2008). In the wake of the 1998 Sissano event, numerous

studies investigated the distribution of submarine landslides along continental slopes, islands and bathymetric highs, as well as their tsunamigenic potential (Canals et al., 2004; Masson et al., 2006; McAdoo et al., 2000; Chaytor et al., 2009; Twichell et al., 2009; Mulder et al., 2009; Tappin, 2010; Iacono et al., 2014; Urgeles and Camerlenghi, 2013; Macías et al., 2015; Palomino et al., 2016; Rodriguez et al., 2012, 2013, 2017). Submarine landslide-generated tsunamis display specific characteristics compared to other sources (Trifunac and Todorovska, 2002; Harbitz et al., 2006, 2014). Vertical displacements induced by the slide can be larger according to its volume and its initial acceleration, which may produce wave amplitudes higher than in the case of an earthquake source (Okal and Synolakis, 2003). Landslide motion can last over several minutes to hours, leading to complex patterns of wave interactions that either amplify or attenuate the wave amplitude (Haugen et al., 2005; Harbitz et al., 2006; Ma et al., 2013; Løvholt et al., 2015). For landslide volumes on the order of a few $km^3$, frequency dispersion of the tsunami results in shorter wavelength and faster wave amplitude attenuation, and limits the far-field propagation of the tsunami.

Tsunami hazard along submarine strike-slip faults remains poorly investigated. Indeed, earthquakes along strike-slip faults generate only minor vertical motion of the seafloor, and hence, minor tsunami, amplitude on the order of a few centimeters. However, releasing or restraining bends with steep slopes may take place along strike-slip faults, promoting submarine failures. For instance, the $Mw$ 6.9 Loma Pietra earthquake along the San Andreas Fault excited tsunami up to 40 cm high in the Monteray Bay,which required a landslide as a secondary source (Ma et al., 1991). The lack of investigation of submarine landslides along submarine strike-slip fault may therefore result in under estimation of tsunami hazard in some places.

The North Anatolian Fault (Fig. 11) is a major, $\sim 1200$-km-long continental strike-slip boundary, which is taking up the dextral relative motion between Anatolia and Eurasia at a current rate of $\sim 25$ mm.yr$^{-1}$ (Reilinger et al., 2006, 2010; Perouse et al., 2012; Müller et al., 2013). In the north Aegean domain the rate decreases from $21.2$ mm.yr$^{-1}$ at the Saros Gulf to 7 mm.yr$^{-1}$ at the Sporades archipelago (Müller et al., 2013). The North Anatolian Fault is one of the most active fault system in Europe with several earthquakes above $M_w \sim 7$ recorded from instrumental seismology in the last decades, among which the deadly Izmit and Duzce events in 1999 (Hubert-Ferrari et al., 2000; Bulut et al., 2018). Detailed historical (Altinok et al., 2011) and field studies (Meghraoui et al., 2012) additionally revealed the geological signature of numerous past Holocene earthquakes and tsunamis. In the Marmara Sea, the response of sedimentary systems to the seismic activity has been investigated throughout detailed stratigraphic comparisons between the timing of mass wasting events and the timing of earthquakes (McHugh et al., 2006; Beck et al., 2007; Drab et al., 2012, 2015). Multibeam and seismic reflection data revealed the signature of numerous submarine landslides (Grall et al., 2012, 2013), among which some may have triggered tsunamis including the AD. 1509 event (Hébert et al., 2005; Özeren et al., 2010; Alpar et al., 2001; Yalçıner et al., 2002).

Although numerous tsunami deposits have been identified in the geological record along the northern aegean coastlines (Reicherter et al., 2010; Papadopoulos et al., 2014; Mathes-Schmidt et al., 2009) and despite the numerous cities along the coastline, the tsunamigenic potential of submarine landslides in this area remains only preliminary investigated (Karambas et al., 2012). The North Aegean Trough is a strike-slip basin emplaced along the North Anatolian Fault. Its tectono-sedimentary context is very similar to the Marmara Sea, with splays of the North Anatolian Fault triggering frequent earthquakes up to $M_w \sim 7$ (such as the 24th May 2014 $M_w 6.9$ event in the Gulf of Saros), and locally gas-charged sediments (Papatheodorou et al., 1993). However, only one conspicuous submarine landslide has been identified so far along the edges of the North Aegean

Trough (Lykousis et al., 2002). This submarine landslide emplaced at $40°\ 15'\ N$, $24°\ 52'\ E$, off Thasos Island, somewhere between $5500$ and $7500 - 8500$ years ago, and removed several $\text{km}^3$ of sediments (Lykousis et al., 2002). The slide was triggered at 300-m-depth, less than 50-km away from the surrounding coastlines, i.e. a configuration very similar to the 1998 Sissano event in Papua-New Guinea. For simplicity, this landslide is hereafter referred to as Thasos landslide. The objective of this paper is to investigate whether the Thasos landslide could have been a potential source of tsunami along the Aegean coasts by the mean of numerical modeling of tsunami generation and propagation. A particular emphasis is put on the possible wave interactions related to the complex morphology of the Aegean coastlines, with numerous islands, bays and peninsulas (Fig. 11). We adopt a deterministic approach based on the evaluation of credible scenarios defined and validated on the basis of the similarity between the observed and the modeled landslide extent.

## 2    Geological background of the Northern Aegean Sea and description of the Thasos submarine landslide

### 2.1    Tectonics and Morphology

The geological history of the Aegean Sea takes place in the complex realm of the collision between Africa and Eurasia during the Cenozoic. The Aegean domain used to be a mountain belt (Hellenides) during the Early Cenozoic, which collapsed since the Late Eocene-Early Oligocene, in close relationship with the southwards retreat of the Hellenic trench (Jolivet and Faccenna, 2000; Jolivet and Brun, 2010; Jolivet et al., 2013, 2015; Brun et al., 2016). Since the Late Miocene, strike-slip tectonics along the North Anatolian Fault system accommodates the lateral escape of Anatolia (Armijo et al., 1999; Faccenna et al., 2006; Hubert-Ferrari et al., 2009; Le Pichon et al., 2015; Bulut et al., 2018; Sakellariou and Tsampouraki-Kraounaki, 2018; Ferentinos et al., 2018). In the North Aegean Sea, the North Anatolian Fault system is divided into two main strike-slip systems, shaping the morphology of the seafloor. The age of inception of these strike-slip segments is estimated in the Plio-Pleistocene from ties of a vintage seismic dataset with industrial wells (Beniest et al., 2016).

The first, main fault segment to the north runs along the Saros Gulf, where it forms an en-échelon, negative flower structure system that connects the North Aegean Trough (Fig. 11) (Kurt et al., 2000; McNeill et al., 2004). This segment accommodates dextral shearing at a $\sim 20\ \text{mm.yr}^{-1}$ rate (Le Pichon and Kreemer, 2010; Perouse et al., 2012). It corresponds to the prolongation of the Main Marmara Fault (Roussos and Lyssimachou, 1991; Le Pichon et al., 2001) west of the Dardanelles Strait (Fig. 11). Numerous oblique NW-SE to E-W splays connects the North Anatolian Fault and form a horsetail structure, typical of strike-slip fault terminations. The horsetail structure forms a $\sim$ 150-km-long, $\sim$ 60-km-wide, $\sim$ 1500 m-deep basin, referred as the North Aegean Trough (Papanikolaou et al., 2002; Sakellariou and Tsampouraki-Kraounaki, 2016; Sakellariou et al., 2016), where the oblique splays isolate a series of half-grabens, tilted over a crustal detachment at $\sim$ 10-km-depth (Laigle et al., 2000). This complex structural pattern results in a very uneven bathymetry, with a series of bathymetric highs and lows within the North Aegean Trough. Southwards, the second strike-slip splay corresponds to the southern branch of the North Anatolian Fault, active at a $\sim 5\ \text{mm.yr}^{-1}$ rate (Le Pichon and Kreemer, 2010; Le Pichon et al., 2014). This structure ends off Skyros Island, where it forms the Edremit Trough (Fig. 11).

## 2.2 Sedimentology

During the Quaternary, the deposition of detritic sediments coming from the surrounding lands (Suc et al., 2015) has been controlled by fast subsidence (0.3 to 1.5 m.Kyrs$^{-1}$) (Piper and Perissoratis, 1991) and glacio-eustatic variations of sea-level (İşler et al., 2008). In the vicinity of the North Aegean Trough, the Late Quaternary (i.e. last 150 kyrs) sediments are composed of

pro-delta formations, composed of sandy to silty turbidites (Piper and Perissoratis, 1991; Lykousis et al., 2002).The uppermost sediments (2 m below seafloor) consist in normally consolidated or slightly over-consolidated muddy to silty sediments, with a bulk density of sediments ranging between 1.4 and 1.5 g.cm$^{-3}$ (Lykousis et al., 2002).

## 2.3 The Holocene Thasos submarine landslide

On the bathymetry, the slide of around 30-km-long scar encloses an area of 85-km$^2$. Downslope, a $75 - m$-thick lobe-shape

deposit spans an area of $\sim 50$ km$^2$ (Fig. 12(a,b))(Lykousis et al., 2002). The mass transport deposit is characterized by a hummocky facies on the seafloor and displays a typical facies composed of chaotic and hyperbolic reflectors on the seismic profiles (Unit 2, in orange on Fig. 12(b,c)). A second 5-m-thick minor mass transport deposit spreads around the failure plan over 8.7-km from the top the scarp (Unit 1, in red on Fig. 12(b,c)). The initial volume, mobilized during the first step of the failure, is estimated following the method described in Ten Brink et al. (2006). It consists in filling-in the failed area according

to the adjacent scar height. The difference between the filled-in grid and the grid displaying the slide scar gives an initial failed volume at 1.85 km$^3$ in the case of the Thasos slide. The volume of the lobe-shaped mass transport deposit lying at the bottom of the scar is estimated at $\sim 3.8$ km$^3$ (Lykousis et al., 2002). The difference between the initial failed volume and the final volume of the slide means that during its flow, the slide has captured sediments from the adjacent slope. The slide evolved as a translational slide of well-bedded sediments, over a glide plane corresponding to a muddy weak layer deposited $170 - 245$

kyrs ago (Lykousis et al., 2002). Although earthquake shaking related to the activity of the North Anatolian Fault may be considered as the most likely trigger of submarine landslides, recent $M_w \sim 7$ events did not trigger significant landslides and tsunami, despite the unstable state of the North Aegean Trough slopes (Lykousis et al., 2002). This discrepancy between strong earthquakes and landslide frequencies has been highlighted in various tectono-sedimentary contexts (Völker et al., 2011; Strozyk et al., 2010; Pope et al., 2015, 2016). Various process may explain this discrepancy, including the amount and the nature

of sedimentary supply, local variations in physical properties of the sediments (Lafuerza et al., 2012) or the over-compaction of the sediments subsequent to the fluid release induced by the seismic wave shaking (Hampton et al., 1996; Strozyk et al., 2010). Some mass transport deposits, probably older than Holocene, can be guessed on sparker lines in the Sporadhes basin (Brooks and Ferentinos, 1980; Ferentinos et al., 1981; Sakellariou et al., 2018; Rodriguez et al., 2018) and on sub-bottom profiles crossing the Saros Gulf (McNeill et al., 2004). However, a comprehensive map of the spatial distribution of landslides

and precise stratigraphic constraints on their recurrence are lacking to further describe how the sedimentary system reacts to the seismic activity of the North Anatolian Fault.

# 3 Numerical modeling of the submarine landslide and the associated tsunami

## 3.1 Bathymetric dataset

The bathymetric grid used for tsunami calculations is a compilation of several grids of different resolutions. We use the multi-beam grid previously published in Papanikolaou et al. (2002), acquired with a seabeam 2120 working at 20 kHz. The horizontal resolution is $\sim 100$ m, for a vertical resolution on the order of 1 m. At the slide location, we include the high-resolution (30-m horizontal resolution) grid acquired by HCMR in 2013-2015 onboard the Aegeo, in the frame of the YPOTHER project (Sakellariou et al., 2016). The remaining gaps have been filled-in with the SRTM PLUS bathymetry (Becker et al., 2009).

## 3.2 Slide modeling: physical background and limits

We use the AVALANCHE code that has been successfully tested from comparisons with tidal measurements for the 1998 Papua New Guinea tsunami (Heinrich et al., 2000) and the 1979 Nice event (Labbé et al., 2012). Two approaches are commonly used to model the dynamics of a submarine landslide. A first approach assimilates the slide to a viscous flow, with a Bingham rheology. The slide is divided into a bottom layer submitted to shear along the glide plane, and a top plug layer with a uniform velocity profile (Jiang and LeBlond, 1993). This approach efficiently reproduced some landslides in the Mediterranean Sea (Iglesias et al., 2012; Løvholt et al., 2015). However, in our case, all the simulations carried for a reasonable range of dynamic viscosities ($25 - 500$ m$^2$.s$^{-1}$) failed to reproduce the first-order morphology of the Thasos slide, and lead to excessive runout and volume values. The second approach, which assimilates the propagation of the landslide to a granular flow (Savage and Hutter, 1989), produced more convincing results in our case, which may be consistent with the detritic nature of the sediments off Thasos (Piper and Perissoratis, 1991; Lykousis et al., 2002). Modelling the slide as a granular flow implies the sediment loses its cohesion immediately after failure occurred at the glide plane. This model assumes that the energy dissipated at the glide plane is much higher than the energy dissipated within the sedimentary flow itself. Although large deformations throughout the thickness of the flow are inevitable (ten Brink et al., 2014) and may have important effects on the evolution of the tsunami wave (Ma et al., 2013), the configuration of the slide (thickness smaller than its length and its width) allows the use of the shallow water assumption (Savage and Hutter, 1989), and therefore the simplification of the mechanical behavior within the sedimentary flow. Deformations within the flow in the earliest stages of collapse are also neglected, as well as the added mass effect (Løvholt et al., 2015), which may result in slightly overestimated initial acceleration (on the order of $10^{-2}$ m.s$^{-2}$). The velocity of the flow in the direction parallel to the slope is considered constant over the entire thickness of the slide. Basal friction at the glide plane is modeled by a Coulomb-type friction law (Savage and Hutter, 1989). This law assumes a constant ratio of the shear stress to the normal stress at the base of the slide and involves a dynamic friction angle $\phi$ between the rough bed and the sliding mass. Granular flows commonly occur for internal friction angles ($\phi$) around $30 - 40$ °. However, this range of values is not appropriate for submarine landslides, which involve $\phi$ values $< 15°$ (Canals et al., 2004; Ten Brink et al., 2009), and even $\phi$ values $< 2°$ for low slope gradient (Rodriguez et al., 2012; Urlaub et al., 2015). Therefore, we perform simulations only for friction angles ranging between 1 and 5° to simulate the Thasos landslide, which spans the range of slope

values in this area. The nonlinear and nondispersive equations from shallow water assumption, written in a coordinate system linked to the topography, give us (Heinrich et al., 2000):

fluid mass conservation:

$$\frac{\partial h}{\partial t} + \frac{\partial}{\partial x}(hu) + \frac{\partial}{\partial y}(hv) = e_w(u+v) \tag{1}$$

momentum conservation equations:

$$\frac{\partial}{\partial t}(hu) + \alpha\frac{\partial}{\partial x}(hu \cdot u) + \alpha\frac{\partial}{\partial y}(hu \cdot v) = -\frac{1}{2}\kappa\frac{\partial}{\partial x}(gh^2\cos\theta) + \kappa gh\sin\theta_x - \tau_{xz(z=0)} \tag{2}$$

$$\frac{\partial}{\partial t}(hv) + \alpha\frac{\partial}{\partial x}(hv \cdot u) + \alpha\frac{\partial}{\partial y}(hv \cdot v) = -\frac{1}{2}\kappa\frac{\partial}{\partial y}(gh^2\cos\theta) + \kappa gh\sin\theta_y - \tau_{yz(z=0)} \tag{3}$$

sediment mass conservation:

$$\frac{\partial}{\partial t}(ch) + \frac{\partial}{\partial x}(chu) + \frac{\partial}{\partial y}(chv) = 0 \tag{4}$$

where:

- $h(x,y,t)$ the thickness of the landslide perpendicular to the glide plane

- $\boldsymbol{u}$ the veloctiy vector with its components $(u,v)$ in accordance with $x$ and $y$ axis, respectively

- $\kappa = 1 - \rho_w/\rho_s$ with $\rho_s$ the sediment density; $\rho_s = 1450$ kg.m$^{-3}$ according to in-situ measurements (Lykousis et al., 2002) and $\rho_w$, the water density. Here we consider $\rho_w = 1000$ kg.m$^{-3}$

- $c$ the sediment concentration at a mean depth

- $\boldsymbol{\theta}(x,y)$ the local sliding angle with $\theta_x$, $\theta_y$ respectively its components in accordance with $x$ and $y$ axis

- $\tau_{xz(z=0)}$ and $\tau_{yz(z=0)}$ are the shear stresses at the bed surface. The constitutive law governing granular flows is generally the Coulomb-type friction law defined in the x-direction by : $\tau_{xz(z=0)} = \kappa gh\cos\theta\tan\phi\frac{u}{\|\boldsymbol{u}\|}$

- $e_w$ the water entrainment coefficient ($e_w \in [0, 0.01]$) (Fukushima et al., 1985; Kostic and Parker, 2006; Sequeiros et al., 2009). It corresponds to the dilution of the sedimentary mass by water incorporation at the interface between the slide and the water.

- $\alpha$ is a parameter characterizing the deviation of the velocity profile compared to a homogeneous distribution profile. Modelling landslide as a granular flow requires a linear profile and $\alpha = 4/3$ (Heinrich et al., 2001).

The main limit of the simulation of a past submarine landslide is the impossibility to provide constraints on the pattern of velocity and acceleration of the slide from the geological record (Harbitz et al., 2006; Ma et al., 2013; Løvholt et al., 2015). We therefore consider scenarios where slide velocities are compatible with the few constraints available from submarine cable breaks recorded in the 20th century i.e., with mean velocity $< 20-30$ m.s$^{-1}$ and initial acceleration $< 0.6$ m.s$^{-2}$ (Heezen and Ewing, 1952; Mulder et al., 1997; Fine et al., 2005).

### 3.3 Tsunami modeling

Simulations of the tsunami waves are also based on the shallow-water approximation, which deals with the full interaction of landslide and water, including the deformation of the sediment body. Equations governing the landslide and the tsunami propagation are similar and are thus solved using the same Godunov-type scheme, extended to second order by using the concept of Vanleer (Alcrudo and Garcia-Navarro, 1993; Mangeney et al., 2000). This model is particularly adapted to nonlinear waves. The time history of sea bottom deformation resulting from the landslide is introduced as a known forcing term $(\cos\theta)^{-1}\partial h/\partial t$ in the mass conservation equation of the tsunami model (Heinrich et al., 2000).

## 4 Results

### 4.1 Elaboration of Landslide and Tsunami Scenarios

We simulate the tsunami that would be triggered in the present-day context by a slide similar to the Thasos slide. Credible scenarios are defined according to the range of parameters able to reproduce as closely as possible the first-order morphology of the observed mass transport deposit (i.e. fitting the following criteria: area of the deposit, volume, runout). Credible scenarios (Table. 11) are obtained for a range of basal friction angle $\phi$ between $1.5-1.8°$ and an initial volume of failure of $1.85$ km$^3$ (Fig. 13). The selection of the most credible scenarios is buttressed on the basis of normalized inverse distance weighting (Shepard, 1968) (Appendix A).

### 4.2 Propagation of the Tsunami potentially generated by the Thasos landslide

For each selected scenario, we present maps of tsunami propagation at different, representative time steps (10, 16, 28 and 43 minutes ; Fig. 14, Fig. 17). We also plot maps of the maximal water heights reached after a $2$ h propagation time in the entire study area (Fig. 15, Fig. 18), which helps to identify bathymetric forcing effect on the tsunami elevation. The propagation of the tsunami at the source displays the same pattern in every scenario, typical of landslide generated tsunami (Ward, 2001; Mohammed and Fritz, 2012; Ma et al., 2013). At the source, the incipient radiated wave has two peaks and one trough (Fig. 14). The water ahead of the front face of the slide (i.e. the out-going wave) is pushed away, creating a leading positive wave in the slide direction (i.e. towards the Chalkidiki Peninsula and the Thermaikos Gulf). The trough following the crest is simultaneously created by the slide excavation, and is followed by a large second positive peak created by the infilling of the trough. In contrast, the front of the wave propagating in the opposite direction of the slide (i.e. the back-going wave) forms a trough propagating towards Alexandroupolis, followed by a second positive wave (Fig. 14). Among the 40 simulations carried out, we present two of them where the deposit is well reproduced, but leading to different tsunami elevation (Table. 12).

#### 4.2.1 First scenario

The first scenario is obtained for a basal friction angle $\phi$ of $1°$ and a water entrainment coefficient of $10^{-3}$. Although the runout of the modeled slide is slightly overestimated by $\sim 2$ km, the modeled volume is $3.7$ km$^3$, which compares closely

to the observations (Fig. 13). The thickness of the modeled deposit, between $20$ and $40$ m, is also in fair agreement with the observations from the seismic line crossing the Thasos slide (Fig. 13). In this scenario, the maximal velocity of the slide reaches the value of $30$ m.s$^{-1}$ about $6$ minutes after the slide beginning. During this phase, the acceleration of the slide is on the order of $0.2$ m.s$^{-2}$. We describe the propagation of the tsunami following the timing of arrival of the leading wave at the coastline.

At the source, the slide generates a tsunami with a maximal water elevation at $3.6$ m (Fig. 15). After $8$ minutes of propagation, the leading out-going wave reaches the Chalkidiki Peninsula and the island of Lemnos, with local amplification up to $1.25$ m. The front of the out-going wave then sweeps the coast of the Athos Peninsula towards the Orfani Gulf. There, the leading wave reaches an elevation up to $1.20$ m-high after $\sim 40$ minutes of propagation (Fig. 14). The tsunami then propagates into the multiple gulf formed by the 'finger-like' pattern of the Chalkidiki Peninsula (Fig. 14). The tsunami enters the Sigitikos Gulf

after $\sim 15$ minutes of propagation, then the Kassandra Gulf after $\sim 28$ minutes of propagation and eventually the Thermaikos Gulf after $\sim 40$ minutes of propagation. Tsunami elevation reaches $0.8 - 0.9$ m in the Sigitikos Gulf, but remains below $0.4$-m in the Kassandra Gulf (Fig. 14). The coastline running from the Sporades to the Thermaikos Gulf is stroke by a $\sim 0.5$-m-high tsunami (Fig. 14). A bathymetric through related to a splay of the North Aegean Trough off Kassandra Peninsula is responsible of the focalization and the amplification of the tsunami before it enters the Thermaikos Gulf (Fig. 14). South of the source,

the leading out-going wave reaches the North of Lemnos after $\sim 10$ minutes of propagation and Agios Efstratios island at $t = 26$ min (Fig. 14), where complex phenomenons of wave interactions lead to tsunami elevation on the order of $1.4$ m at the coastline. The leading back-going wave is split in several fronts propagating at different speeds when it reaches the Samothraki and Gokceada Islands after $\sim 16$ minutes of propagation. A $1.65$-m-high tsunami strikes Alexandroupolis after 1h07min of propagation (Fig. 14). The complex pattern of the Aegean coastline results in a complex pattern of wave interactions that either

destroy or amplify the tsunami (Fig. 16). Amplification through resonance is expected within the Sigitikos Bay, the Orfani Gulf, and the Pournias Bay north of Lemnos (Fig. 14). South East of Agios Efstratios and north of Samothraki, we observe lineaments where the tsunami wave is amplified (on the order of $50$ cm), despite the lack of bathymetric feature that could explain local forcing. Constructive interferences between the waves that surrounded these islands may produce this amplification. At both locations, the leading wave is split in two wave fronts that rotate along the islands by a coastal attraction phenomenon, until the

two wave fronts merge together, leading to wave amplification. For instance, the back-going wave of the tsunami is split in two wave fronts when it strikes Samothraki at $t = 28$ min (Fig. 16). After their propagation around the island, both fronts collide at $t = 36$ min, which results in constructive interferences that amplify the wave up to $\sim 0.5 - 0.6$ m. A second positive wave front, which has been amplified along the Thracian coast (Fig. 16 at $43$ min), collides with the back-going wave front formed east of Samothraki. This interference cancels the negative polarity of the back-going wave, and even amplifies the tsunami

front up to $1.65$ m at Alexandroupolis.

### 4.2.2   Second scenario

The second selected scenario is obtained for a basal friction angle $\phi$ of $1.5°$ and a water entrainment coefficient $e_w$ of $5.10^{-4}$. In this model, the area of the mass transport deposit is larger than the observations, resulting in an overestimated but still reasonable volume of $4.1$ km$^3$ (Fig. 13). In this case, the maximal velocity of the slide reaches $25$ m.s$^{-1}$ about $3$ minutes

afters the slide triggering. During this phase, the slide acceleration is on the order of $\sim 0.15$ m.s$^{-2}$. This second scenario leads to the same timing and mode of propagation of the tsunami, with similar processes of wave amplifications and interactions around the islands of Agios Efstratios and Samothraki (Fig. 17). However, the tsunami elevation is less important everywhere. The maximum of water elevation is reduced at $\sim 85$ cm along the Chalkidiki Peninsula, at $\sim 1.10$ m at Alexandroupolis, and around 50 cm at Agios Efstratios and north of Lemnos. Amplification within the bay of Sigitikos, Kassandra, and the Gulf of Thermaikos is also reduced, with water heights less than 20 cm (Fig. 18).

## 5   Discussion

Our simulations were able to strikingly reproduce the first-order morphological characteristics of the Thassos slide, using basic granular flow behaviour governed by a simple Coulomb friction law at the glide plane. Therefore, the selected range of physical parameters used for the granular flow modeling, and the resulting tsunami, are considered realistic. Compared to previous studies (Karambas et al., 2012), the advantage of our approach is the coupling between the dynamic of the slide, the formation of the tsunami and its propagation.

The results of our simulations show that the Thasos slide has probably been tsunamigenic. If a similar slide occurs at the current sea level, the areas of Alexandroupolis (57812 inhabitant in 2011 according to the Hellenic Statistical Authority), the Chalkidiki Peninsula and Agios Efstratios would be the most threatened, with tsunami elevations at the coastline ranging between 1 and 2 m according to the model parameters.

The Tsunami strikes most of the coastlines of the northern Aegean Sea after less than 1 hour of propagation (Table. 12). The Athos Peninsula is reached after just 10 minutes, but is not densely populated and the topography does not allow large run up distances. The geomorphological context of the Thassos Slide is similar to slide at the origin of the Papua-New Guinea tsunami in 1998 (roughly the same volume, slope, and distance to the coastline), but the numerous interactions between waves due to the numerous bays and islands makes the expected high of the tsunami at the Aegean coastline lower. The tsunami simulations have some limits. The first one is the lack of high-resolution bathymetry along the coast, which does not allow accurate computation of the flooding onland. It remains unknown if local bathymetric features or harbors would result in further amplification or dispersion of the tsunami wave. This is especially critical for the numerous gulf (Orfani, Sigitikos and Kassandra) where resonance is expected. Run-up values being often more important than the water height calculated a few kilometers from the coastline (Synolakis, 1987), the computed tsunami elevations at the coastline should be considered as minimum values of the actual run up. Moreover, the initial slide acceleration and its speed strongly impinge on the initial pattern of the tsunami, including the pattern of short wavelengths, and therefore on the subsequent frequency dispersion (Haugen et al., 2005). Even if the modeled values of slide velocity and acceleration fit with values measured on slides triggered in the 20th-21st centuries (Heezen and Ewing, 1952; Mulder et al., 1997; Pope et al., 2016), geological records are not sufficient to determine these parameters for Holocene period.

A second-order source of simplification is the non-dispersive numerical model that is not able to account for frequency dispersion inherent in tsunamis with short wavelengths. However, the density of the mobilized sediments is quite low ($\sim 1500$ kg.m$^{-3}$) . Frequency dispersion at the source is expected to be negligible for this range of densities (Ma et al., 2013).

## 6 Conclusion

The results of our simulations show that the expected tsunami wave from the Thasos slide are higher (165 cm for Alexan-droupolis or 145 cm for the costs of Agios Efstratios, Sporades Archipelago and Athos Peninsula) than values expected in the case of an earthquake along the North Anatolian Fault, and higher than run up values (between 20 and 50 cm south of Thes-saloniki) documented from tsunami deposits in the Thermaikos Gulf (Reicherter et al., 2010). The study highlights the need to build a comprehensive map of the distribution of landslides within the North Aegean Trough, as well as a full quantification of their volumes, to better estimate the variety of tsunami scenarios in the area . This discrepancy between earthquake and land-slide recurrence asks the question of the response of sedimentary systems to ground shaking. The tsunami hazard related to submarine landslides similar to Thassos is less severe than tsunami associated to landslides triggered during the 1956 Amorgos event (Perissoratis and Papadopoulos, 1999; Okal et al., 2009); or tsunami triggered during or subsequent to volcanic events (e.g. Santorini in the Bronze age (Novikova et al., 2011).

### Appendix A:  Selection of Tsunami scenario

In order to quantify the gap between landslide simulation results and data from bathymetry and seismic profiles we used an 'normalized inverse distance weighting' function. Thereby we define a weight function $\omega$ as:

$$\omega = \sum_{k=1}^{4} \frac{1}{d_k(x_k, x_k^{th})^p} \tag{A1}$$

Where:

- $x_k$ are parameters used for determined the differences between the numerical simulation and the bathymetry. We used for $x_k$: (1) the sliding direction, (2) the total surface of deposit, (3) the amplitude of the discontinuity in the thickness of sediments between units 1 and 2 (Fig. 12) and (4) the final volume of the deposit.

- $x_k^{th}$ are the used observable parameters from the bathymetry and the seismic profiles corresponding to $x_k$

- $d_k(x_k, x_k^{th})$ are distance functions between $x_k$ and $x_k^{th}$. We define here $d_k$ as $d_k = |x_k^{th} - x_k|$

- $p$ is a real positive number, called the power parameter. If $p > 1$, the contribution in $\omega$ of the small distance is amplified (we used here $p = 1$).

25 The higher the $\omega$, the closer the scenario is to bathymetry and seismic data. The function $\omega$ is calculated for all scenarios and allows us to highlight the two scenarios presented in this study. The computation of $\omega$ for the 2 credible scenarios gives $\omega = 181.52$ and $\omega = 2.56$ (Fig. 13)

*Acknowledgements.* We acknowledge the support of the "Yves Rocard" Joint Laboratory between the École Normale Supérieure, the Comissariat à l'Énergie Atomique, and the CNRS.

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

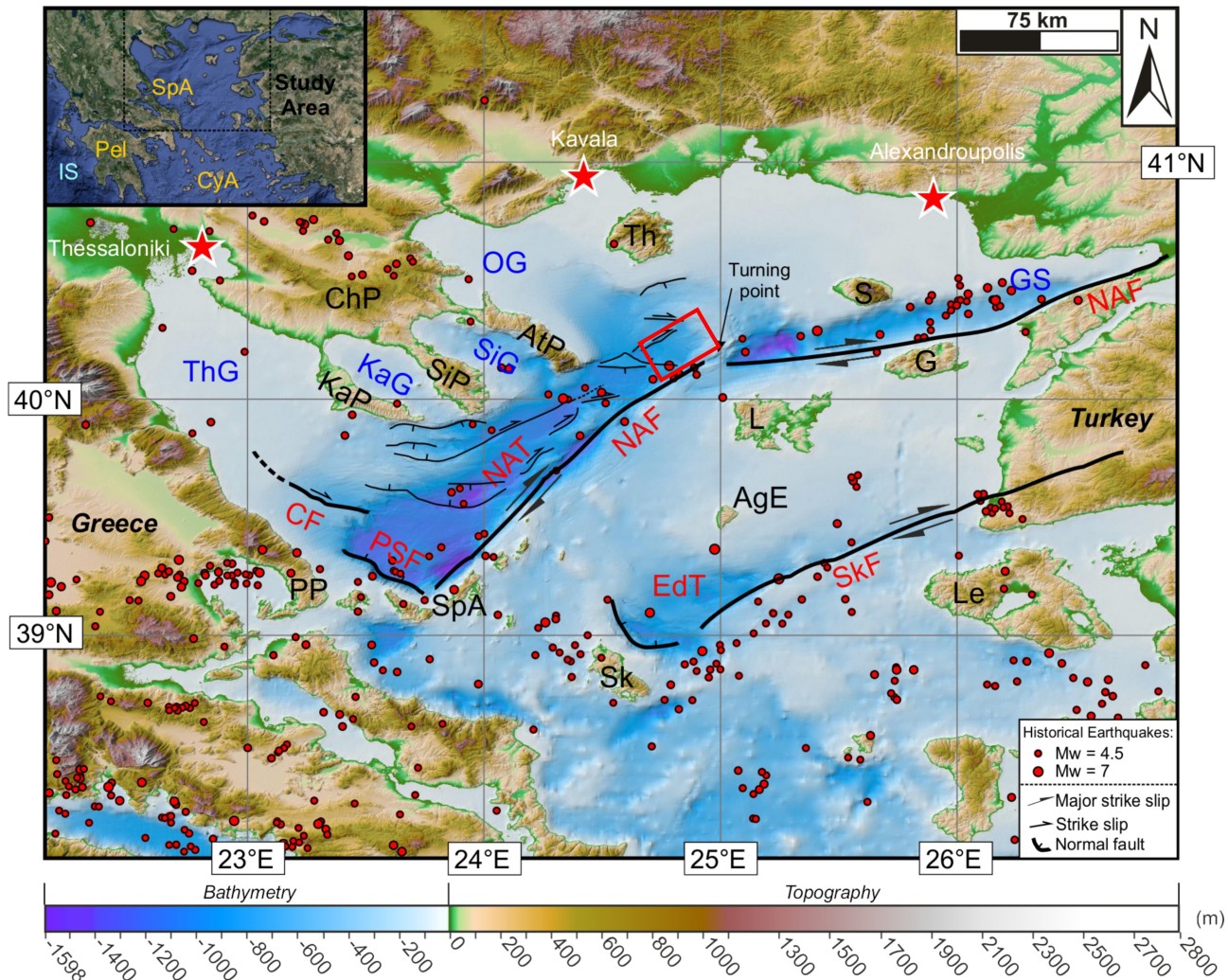

**Figure 11.** General morphological features of the studied area. The main fault are indicated (after Sakellariou and Tsampouraki-Kraounaki (2016)). Seismicity from USGS, since January 1st 1950, for earthquake with $M_w > 4.5$. The red rectangle gives the location of the Thasos landslide. In inset, location of the study area (background from Google Earth). AgE: Agios Efstratios, AS: Aegean Sea, AtP: Athos Peninsula, ChP: Chalkidiki Peninsula, CyA: Cyclades Archipelago, EdT: Edremit Trough, G: Gokceada (Imvros), GS: Gulf of Saros, IS: Ionian Sea, KaG: Kassandra Gulf, KaP: Kassandra Peninsula, L: Lemnos, Le: Lesvos, NAT: North Aegean Trough, NAF: North Anatolian Fault, OG: Orfani Gulf, Pel: Peloponnese, PP: Pelion Peninsula, PSF: Pelion-Skopelos Fault, S: Samothraki, SiG: Sigitikos Gulf, SiP: Sithonia Peninsula, Sk: Skyros, SkF: Skyros Fault, SpA: Sporades Archipelago, Th: Thasos, ThG: Thermaikos Gulf. General topography and bathymetry from SRTM30 (Becker et al., 2009) and high resolution bathymetry ($250m$) from Sakellariou and Tsampouraki-Kraounaki (2016).

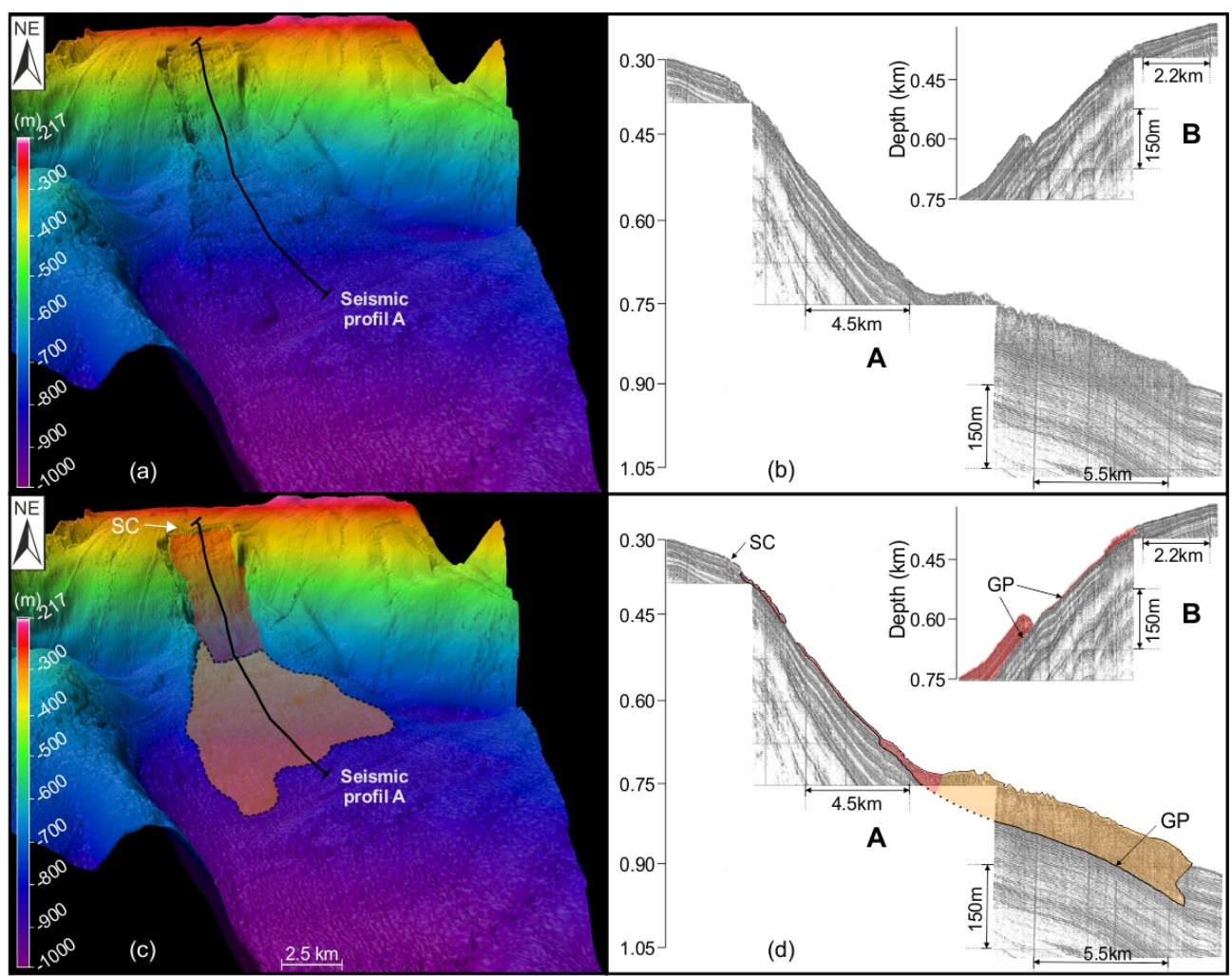

**Figure 12.** Features of the Thasos submarine landslide. In red, the first unit, in orange, the second unit. (a): 3-D view of the actual bathymetry of the Thasos landslide. (b): raw seismic profiles from Lykousis et al. (2002): (A) Downslope air-gun sub-bottom profile indicated on (a) and (B) details of the seismic profile along the major glide zone. (c): interpretation of the 3-D bathymetry : in red the area where the deposit is thin or non-existent in the upper part of the slide; in orange the area where the deposit is thick (marqued on the bathymetry by a hummocky facies and in the seismic profile by chaotic and hyperbolic reflectors); SC: slide scarp. (d): Interpretation of seismic profiles with the same convention as in (c), GP: glide plane. (b) and (d): modified after Lykousis et al. (2002).

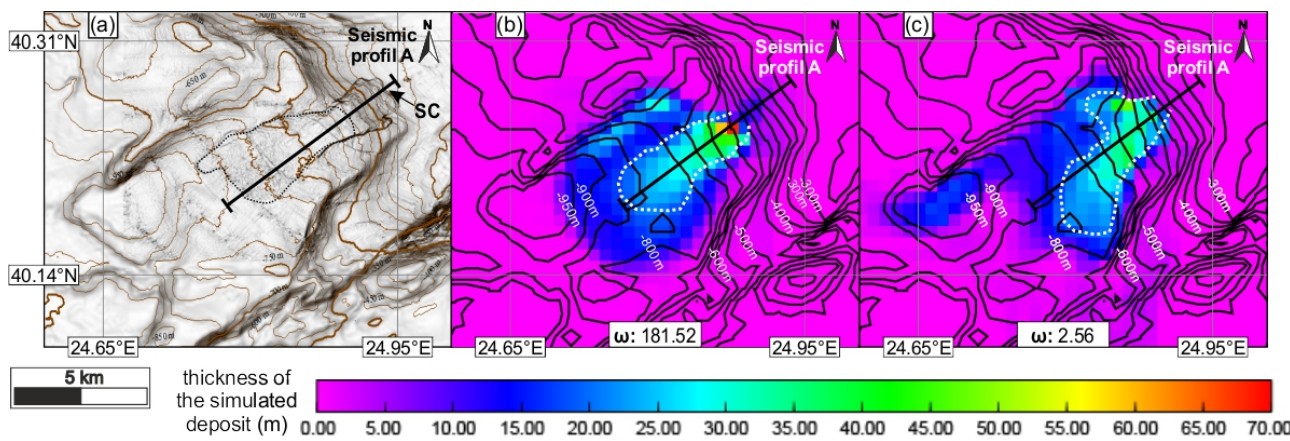

**Figure 13.** Numerical reconstruction of the final position of the slide for the two credible scenarios. (a): Multibeam bathymetry of the slide. Black dotted line corresponds to the limit of the slide deposit (in orange on Fig.12). (b): position of the slide's deposit for the first credible scenario. White dotted line highlights the main transport deposit (above 20 m thick). The resolution of the seismic profile is not enough to track the mass transport deposit thin than 20 m thick. (c): position of the slide's deposit for the second credible scenario with the same convention as in (b). $\omega$ is a coherence index. The higher this index is, the closer the simulation is to the observable data. This index is defined with normalized inverse distance weighting (Appendix A). SC: slide scarp.

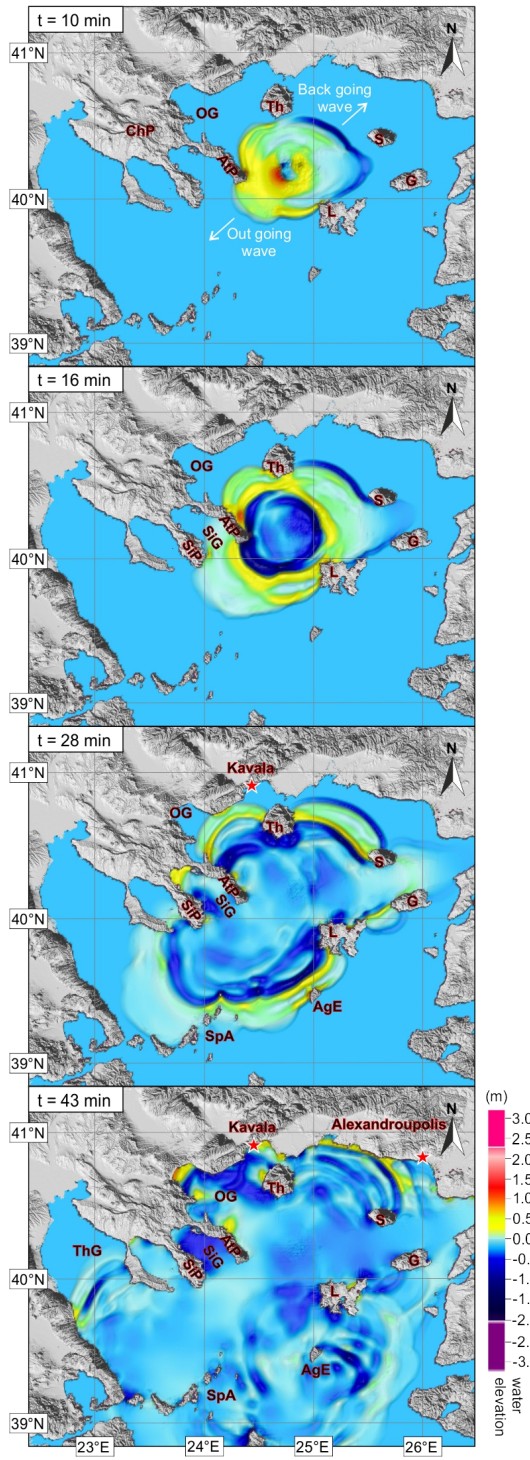

**Figure 14.** Tsunami propagation in the case of the first credible scenario. ($e_w$ of $10^{-3}$ and a basal friction angle of $1°$; Table. 11). AgE: Agios Efstratios, AtP: Athos Peninsula, G: Gokceada, L: Lemnos, OG: Orfani Gulf, S: Samothraki, SiG: Sigitikos Gulf, SiP: Sithonia Peninsula, SpA: Sporades Archipelago, Th: Thasos, ThG: Thermaikos Gulf.

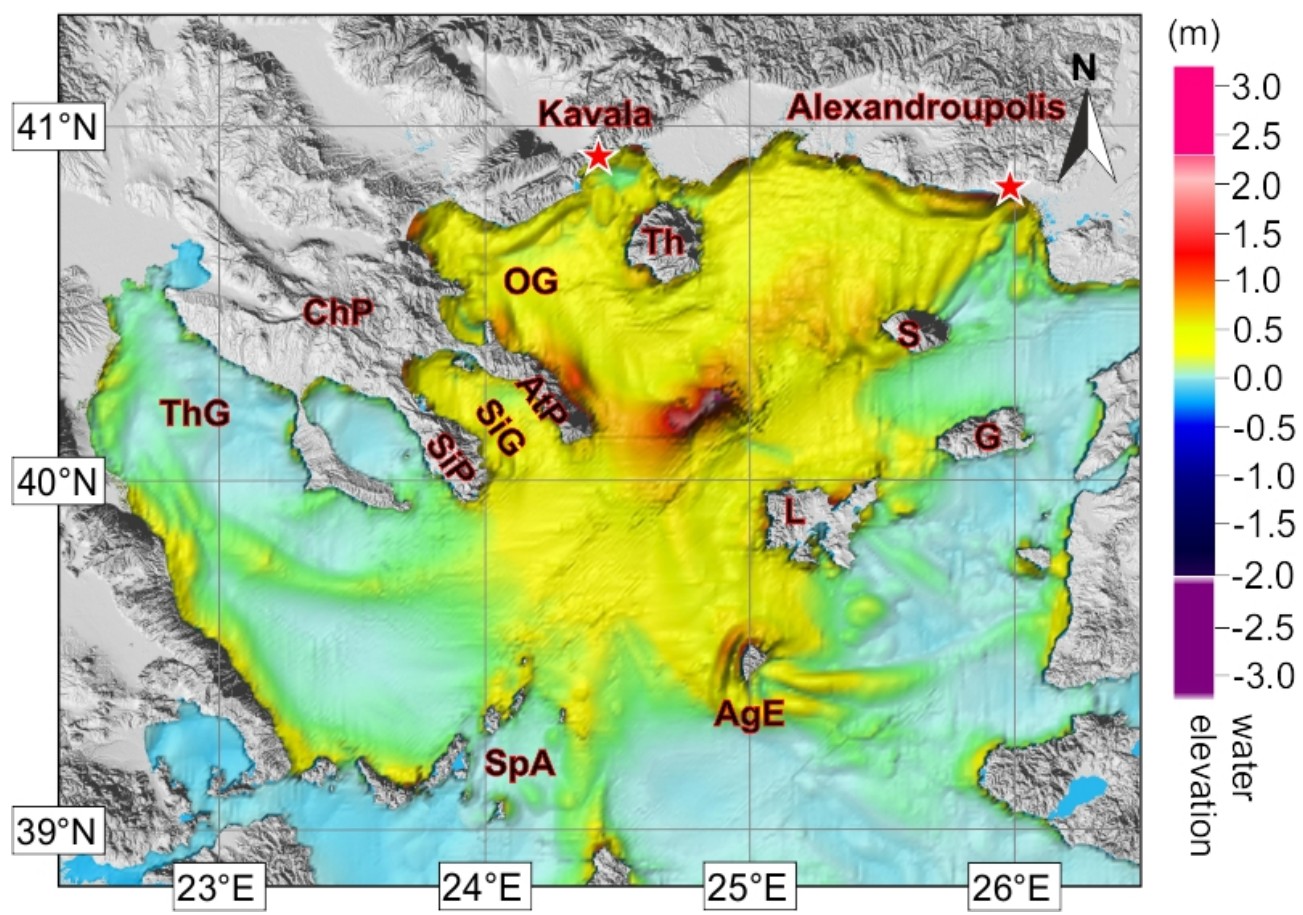

**Figure 15.** Figure of the maximum water elevation of the tsunami for the first credible scenario (Table. 11). AgE: Agios Efstratios, AtP: Athos Peninsula, G: Gokceada, L: Lemnos, OG: Orfani Gulf, S: Samothraki, SiG: Sigitikos Gulf, SiP: Sithonia Peninsula, SpA: Sporades Archipelago, Th: Thasos, ThG: Thermaikos Gulf.

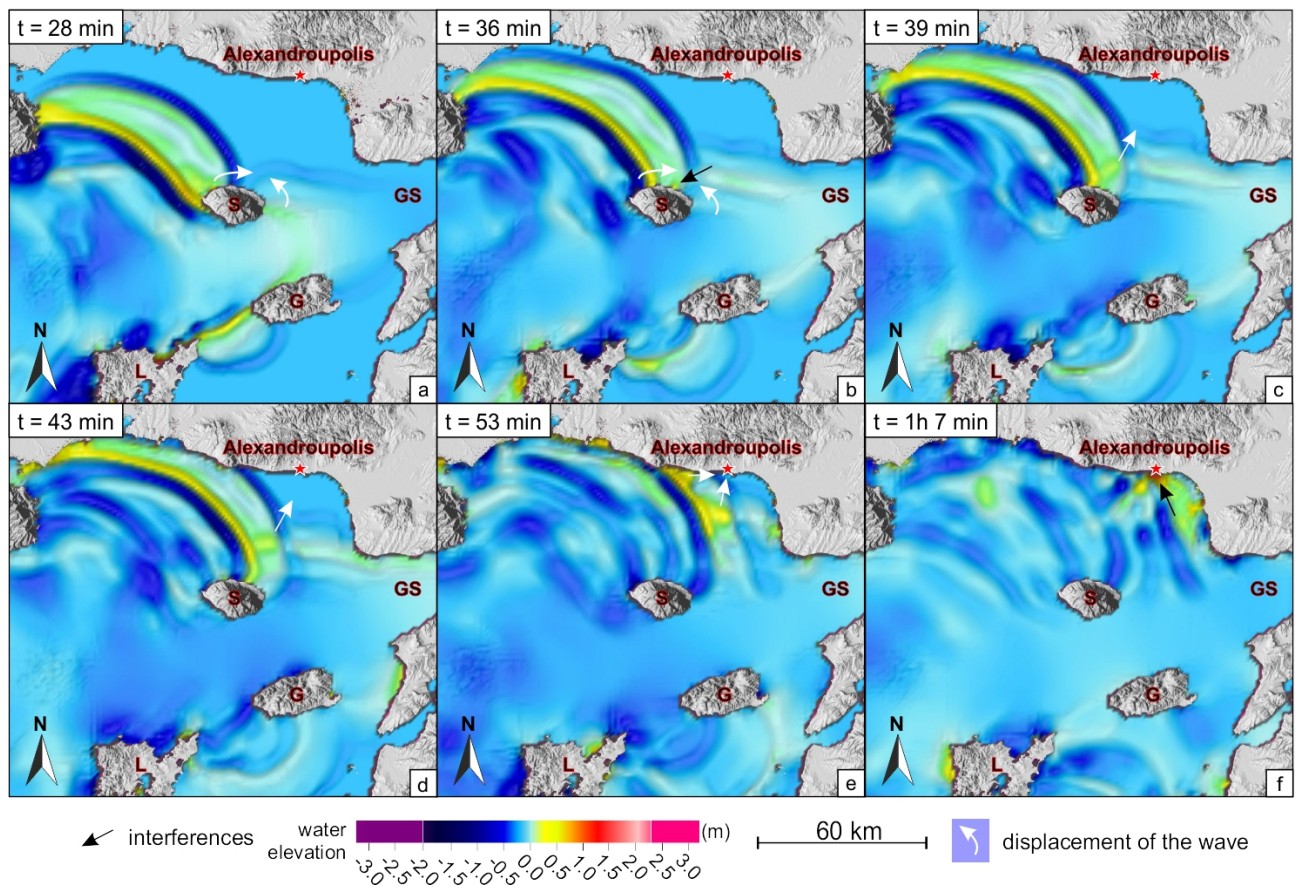

**Figure 16.** Tsunami propagation in the NE of Aegean Sea : illustration of a complex system of amplification by inteferences for the first credible scenario (Table. 11). GS: Gulf of Saros G: Gokceada, L: Lemnos, OG: Orfani Gulf, S: Samothraki. (a): arrival in the South of Samothraki of wavefront, (b): rotation of the wavefront around Samothraki and development of constructive interferences in the North of the island., (c): recess of the wavefront of the island, (d): migration to the North for the new wavefront, (e): arrival of a other front by the west, (f): constructive interferences in front of Alexandroupolis and formation of a wave of $1.65\ m$ high.

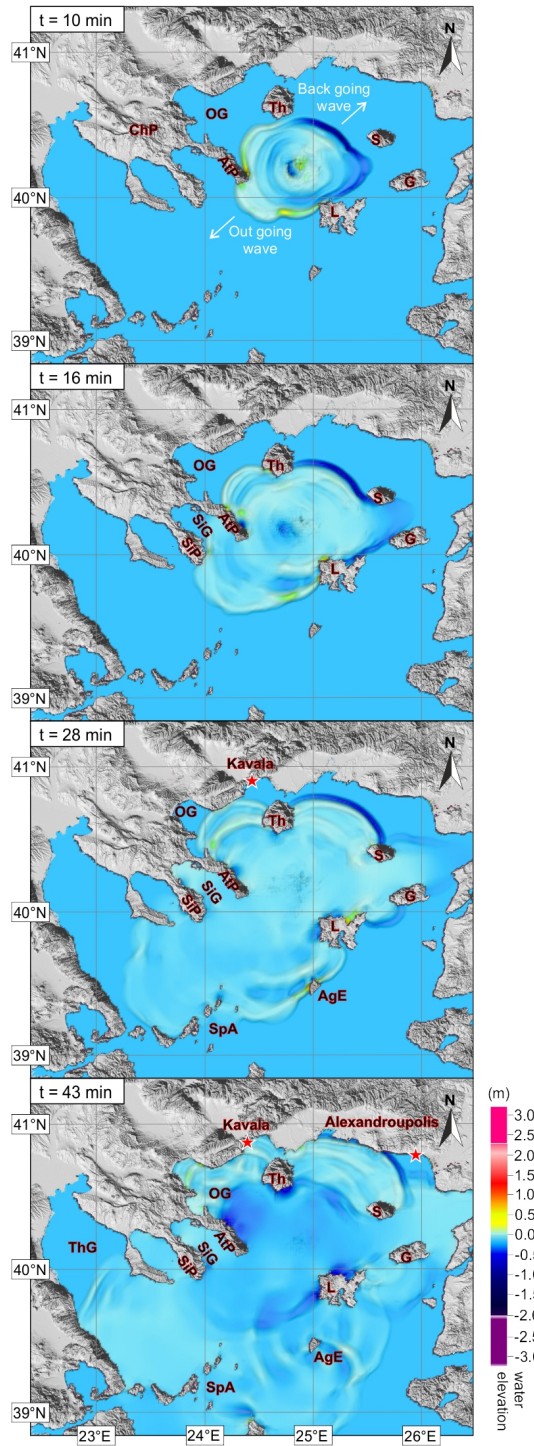

**Figure 17.** Tsunami propagation in the case of the second credible scenario. ($e_w$ of $5 \cdot 10^{-4}$ and a basal friction angle of $1.5°$ ; Table. 11).
AgE: Agios Efstratios, AtP: Athos Peninsula, G: Gokceada, L: Lemnos, OG: Orfani Gulf, S: Samothraki, SiG: Sigitikos Gulf, SiP: Sithonia
Peninsula, SpA: Sporades Archipelago, Th: Thasos, ThG: Thermaikos Gulf.

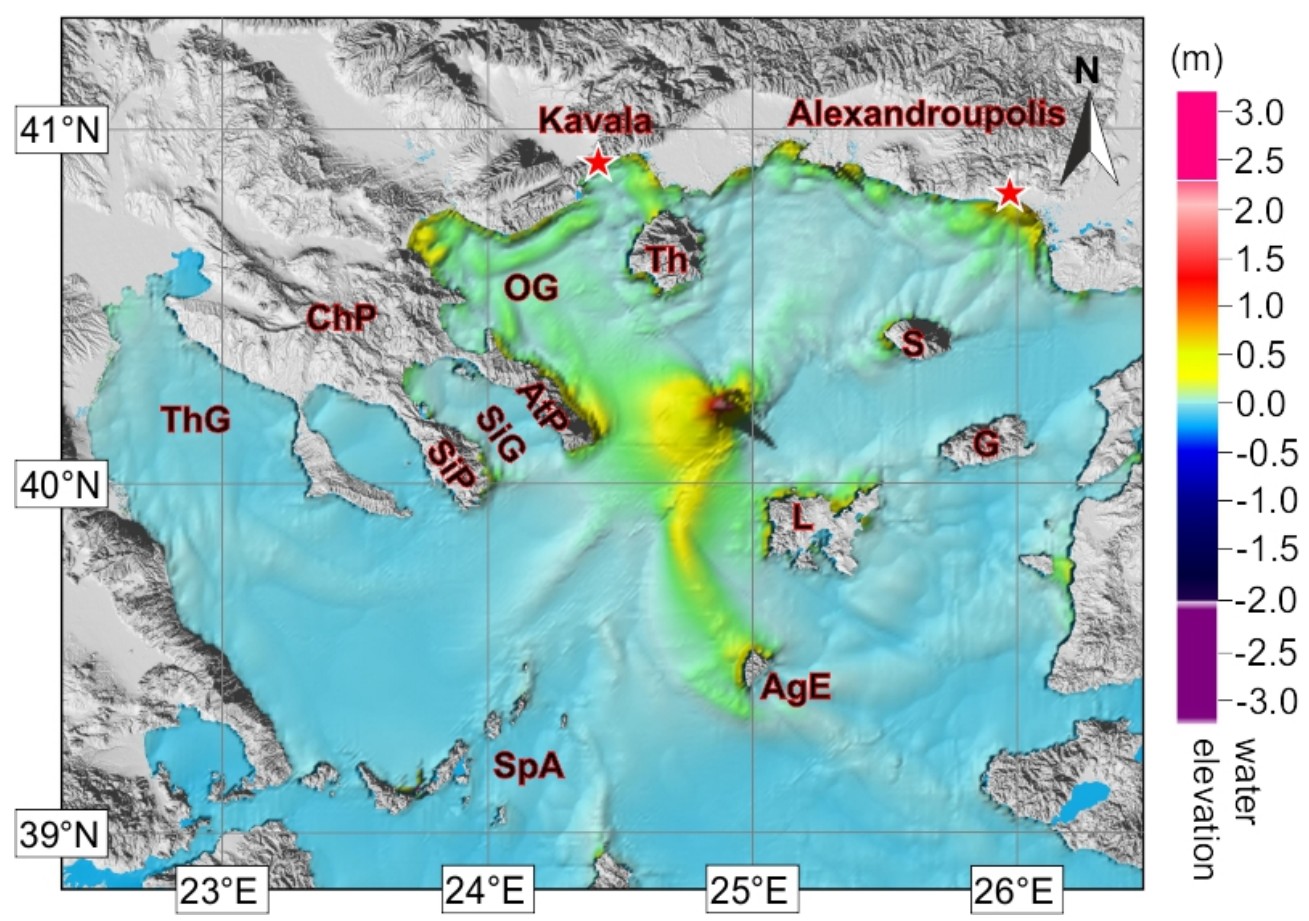

**Figure 18.** Figure of the maximum water elevation of the tsunami for the second credible scenario (Table. 11). AgE: Agios Efstratios, AtP: Athos Peninsula, G: Gokceada, L: Lemnos, OG: Orfani Gulf, S: Samothraki, SiG: Sigitikos Gulf, SiP: Sithonia Peninsula, SpA: Sporades Archipelago, Th: Thasos, ThG: Thermaikos Gulf.

**Table 11.** Table of key parameters used for produced the 2 most credible scenrios. Vi: Initial volume, TRUN: total simulated time, TSLISTOP: slide is stopped at tslistop, PHIBAS: basal friction angle, $e_w$: water entrainment coefficient, Vf: final volume of the slide.

| Credible scenarios | Vi($km^3$) | TRUN(s) | TSLPISTOP(s) | PHIBAS($°$) | $e_w$ | Vf($km^3$) |
|---|---|---|---|---|---|---|
| $1^{st}$ scenario | 1.8511 | 7200 | 600 | 1.0 | $1 \cdot 10^{-3}$ | 3.6938 |
| $2^{nd}$ scenario | 1.8511 | 7200 | 1700 | 1.5 | $5 \cdot 10^{-4}$ | 4.1103 |

**Table 12.** Table of maximum elevation (cm) of the water level for 10 places where the tsunami was amplified. Top: Above the marine landslide, AgE: Agios Efstratios, AtP: Athos Peninsula, L: Lemnos (Gulf of Pournias), OG: Orfani Gulf, SpA: Sporades Archipelago, Th: Thasos.

| Credible scenarios | Top | Th | L | AgE | SpA | AtP | OG | Kavala | Alexandroupolis | Turkey |
|---|---|---|---|---|---|---|---|---|---|---|
| $1^{st}$ scenario | 360 | 115 | 140 | 145 | 145 | 145 | 120 | 65 | 165 | 75 |
| $2^{nd}$ scenario | 470 | 75 | 105 | 110 | 80 | 120 | 40 | 25 | 60 | 40 |