# Peer review of "Tsunamigenic potential of a Holocene submarine landslide along the North Anatolian Fault (North Aegean Sea, off Thasos Island): insights from numerical modeling"

_Natural Hazards and Earth System Sciences, 2017_

## Referee Comment (RC1) · K. Kawamura (Referee) · 11 Jul 2018

This paper deals with a tsunamigenic submarine landslide in the Aegean Sea. The authors discussed credible scenarios of the tsunami propagation by a large slide around the Aegean Sea using topographic and geologic studies and computer simulations. I have a few questions about the topographic and geologic studies as below.

First question: How do you draw the outline of the Thasos submarine landslide in Figure 2 (a) ? We know the slide scar and front of the wasted mass from Figure 2

[Figure]

(a) and (b), but we do not understand the lateral limit of the wasted mass from these dataset. We could see a detailed topography of the slide including the wasted mass region in Figure 3 (a). The authors also draw the outline of the slide in this figure, but we could not discuss about this issue in detail, because the bathymetric map is too small. Please show us clearly the bathymetric map in this article, and please explain how do you identify the slide from these datasets.

Second question: How do you interpret the seismic image in Figure 2 (b)? The authors show us SC, SLD, SLB, SLP, and GP in this figure. The SLP area is painted out an opaque orange color, so that we could not see how you interpret the seismic image. The GP steps down, but we could not check your interpretation. It is because the figure is too small. Please enlarge this figure with indication of transparent color.

This is not question, but please indicate the location of the Thasos submarine landslide in Figure 1.

---

## Author Comment (AC1) · 15 Sep 2018

**Dear editors and reviewer,**

You will find attached to this letter (as supplement) the last corrected version of 'Tsunamigenic potential of a Holocene submarine landslide along the North Anatolian Fault (North Aegean Sea, off Thasos Island): insights from numerical modeling'. We will here answer questions raised by the review process.

Answer to question 1:

I change the figure 2 to show clearly the bathymetry of the Thasos slide (I also add the raw seismic profile). The bottom limit of the landslide have been draw thank to seismic profile (figure 2b and figure 2d) and thank to bathymetry. Indeed the termination of the slide deposit is clearly noticeable on seismic data and this deposit is marked on the bathymetry by surface asperities (hummocky facies on the seafloor as mentioned in the manuscript). I have also added some precisions on the legend of figure 3: The white lines drawn to delimit the slide's deposit correspond to the main transport deposit. However, a thinner deposit (less than 20 m thick) may extend beyond this limit but the resolution of seismic data if not enough to detect it. Overall, the captions of the figure have been simplified.

Answer to question 2:

The notation previously used in Figure 2 (e.i. SC, SLD, SLB, SLP and GP) come from the paper of Lykousis et al., 2012. However, as they are not essential in this manuscript and as we did not re-interpret them, we decide to remove these notations from all figures.

Additional comments:

Thanks to the good advice of the reviewer, we add the location of the Thasos slide on the general map of the north Aegean Basin (Figure 1) and the size of the figures has been increased. Moreover, some minor corrections have been made on the language and the bibliography has been updated to 2018.

We hope this corrected manuscript with fulfill the editor's expectations and that they will find it of great interest for NHESS.

We thank K. Kawanura for his rigorous and usefull work of review.

Best Regards, Alexandre Janin, on behalf of co authors

N.B. The entire corrected manuscript is attached as supplement file.

Please also note the supplement to this comment: https://www.nat-hazards-earth-syst-sci-discuss.net/nhess-2017-405/nhess-2017-405-AC1-supplement.pdf

Fig. 1. Figure 2 corrected

---

## Author Comment (AC2) · 3 Oct 2018

I thank warmly reviewer and editors for their work.

---

## Author Response (AR1)

**Tsunamigenic potential of a Holocene submarine landslide along the North Anatolian Fault (North Aegean Sea, off Thasos Island): insights from numerical modeling**

*A. Janin*[1], *M. Rodriguez*[1], *D. Sakellariou*[2], *V. Lykousis*[2] and *C. Gorini*[3]

[1]Laboratoire de Géologie de l'Ecole normale supérieure de Paris; PSL research university, CNRS UMR 8538, 24 rue Lhomond, 75005 Paris, France.
[2]Institute of Oceanography, Hellenic Center of Marine Research, GR-19013 Anavyssos, Greece
[3]Sorbonne Universités, UPMC Université Paris 06, UMR 7193, ISTeP, F-75005, Paris, France.
* * *
Dear editor,

Here are the answer to your comments. Your comments are in bold, the answers are in regular font, quotations from the revised version of the manuscript are in italics.

**1. A paragraph in the Discussion describing what you have novelty learned from your simulations about the nature of landslide induced tsunamis, globally. Compare to other recent/classic works (the work must show a beyond local implication).**

Landslides with volumes on the order of a few $km^3$ can be well simulated with a granular flow or a Bingham flow- in our case, the granular flow behaviour fits better the observations. This statement has been verified by numerous authors working on the case of the Papua-New Guinea event in 1998. Therefore, our study does not bring anything new on the genesis of landslide tsunami (i.e. in terms of physics). Conversely, one of the strong point of our work is to show that such a simple rheology is able to reproduce the observed characteristics of the landslide.
Current developments on the genesis of landslide tsunamis concern larger slide, with volumes on the order of $> 1000 km^3$ (group of Finn Lovholt in Norway). Moreover, works dedicated to the development of new numerical methods are focused on recent tsunamis, the fit with tide gauge data being the key for the validation of the model. Publications dedicated to the development of new methods of tsunami modelling are usually intended for JGR, GJI or Pure and Applied Geophysics.
Our objective is totally different : we look for potential sources of past tsunamis, for which observations are lacking. This objective is important because even a moderate tsunami can cause dramatic damage if the populations are not aware of the risk. Because earthquakes along strike slip faults only produce small offsets of the seafloor, tsunami hazard along this type of fault is generally under evaluated. The general (i.e. beyond local) objective of our work is to attract the attention on the fact that submarine landslides along strike slip fault may be a source of tsunami. This is not new, but this is very poorly documented so far. Our study show that tsunami hazard along the North Anatolian Fault, the most dangerous fault in Europe, has been under-estimated. We think this result may catch the attention of the NHESS readership.

Modifications of the manuscript:

In the introduction:

'Tsunami hazard along submarine strike-slip faults remains poorly investigated. Indeed, earthquakes along strike-slip faults generate only minor vertical motion of the seafloor, and hence, minor tsunami, amplitude on the order of a few centimeters. However, releasing or restraining bends with steep slopes may take place along strike-slip faults, promoting submarine failures. For instance, the $Mw6.9$ Loma Pietra earthquake along the San Andreas Fault excited tsunami up to $40$ cm high in the Monteray Bay, which required a landslide as a secondary source (Ma et al., 1991). The lack of investigation of submarine landslides along submarine strike-slip fault may therefore result in under estimation of tsunami hazard in some places.'

In the discussion, we summarize better the interest of our approach:

'Our simulations were able to strikingly reproduce the first-order morphological characteristics of the Thassos slide, using basic granular flow behaviour governed by a simple Coulomb friction law at the glide plane. Therefore, the selected range of physical parameters used for the granular flow modelling, and the resulting tsunami, are considered realistic. Compared to previous studies (Karambas et al. 2012), the advantage of our approach is the coupling between the dynamic of the slide, the formation of the tsunami and its propagation.'

**2. Add a Conclusion part (after the Discussion).**

Done.

'Despite these limits, the results of our simulations show that the expected tsunami wave from the Thasos slide are higher ($165$ cm for Alexandroupolis or $145$ cm for the costs of Agios Efstratios, Sporades Archipelago and Athos Peninsula) than values expected in the case of an earthquake along the North Anatolian Fault, and higher than run up values (between $20$ and $50$ cm south of Thessaloniki) documented from tsunami deposits in the Thermaikos Gulf (Reicherter et al., 2010). The study highlights the need to build a comprehensive map of the distribution of landslides within the North Aegean Trough, as well as a full quantification of their volumes, to better estimate the variety of tsunami scenarios in the area. This discrepancy between earthquake and landslide recurrence asks the question of the response of sedimentary systems to ground shaking. The tsunami hazard related to submarine landslides similar to Thassos is less severe than tsunami associated to landslides triggered during the 1956 Amorgos event (Perissoratis and Papadopoulos, 1999; Okal et al., 2009); or tsunami triggered during or subsequent to volcanic events (e.g. Santorini in the Bronze age (Novikova et al., 2011).'

**3. Also please explain what makes the difference in landslide volume $1.85$ and $3.8 km^3$ (P. 4; lines 8 - 9), and how age was determined.**

Modified as follows:

[revised manuscript text omitted]

---

## Author Response (AR2)

**Tsunamigenic potential of a Holocene submarine landslide along the North Anatolian Fault (North Aegean Sea, off Thasos Island): insights from numerical modeling**

*A. Janin[1], M. Rodriguez[1], D. Sakellariou[2], V. Lykousis[2] and C. Gorini[3]*

[1]Laboratoire de Géologie de l'Ecole normale supérieure de Paris; PSL research university, CNRS UMR 8538, 24 rue Lhomond, 75005 Paris, France.
[2]Institute of Oceanography, Hellenic Center of Marine Research, GR-19013 Anavyssos, Greece
[3]Sorbonne Universités, UPMC Université Paris 06, UMR 7193, ISTeP, F-75005, Paris, France.
* * *
December 25, 2018

Dear editor,

I modified my manuscript according to your four minor comments (in bold below). My response is in italics.

- **P. 2, Line 2 - 4: Consider adding Katz et al., 2015 to this reference list**
  *I added this publication to the reference list*

- **P. 2, Line 5 -6: Change to: Vertical displacements induced by the slide can be larger than displacements caused by earthquake related on-fault sea floor rapture and thus may produce higher wave amplitudes (Okal and Synolakis, 2003).**
  *Done*

- **P. 2, Line 12: change 'motion' to 'displacement'.**
  *Done*

- **P. 2, Line 34: 'and locally gas-charged sediments'; This text does not logically connected to the sentence, please modify.**
  *Minor change was enough to clarify the text. The sentence becomes: 'Its tectono-sedimentary context is very similar to the Marmara Sea with splays of the North Anatolian Fault triggering frequent earthquakes up to $M_w \sim 7$ (such as the 24th May 2014 $M_w$6.9 event in the Gulf of Saros) and with locally gas-charged sediments (Papatheodorou et al., 1993).'*

Hoping these modifications will clarify the manuscript,
Sincerely yours,

Alexandre Janin, on behalf of co authors